# LEAF: A LEARNABLE FRONTEND FOR AUDIO CLASSIFICATION

**Neil Zeghidour, Olivier Teboul, Félix de Chaumont Quitry & Marco Tagliasacchi**
Google Research
{neilz, oliviert, fcq, mtagliasacchi}@google.com

## ABSTRACT

Mel-filterbanks are fixed, engineered audio features which emulate human perception and have been used through the history of audio understanding up to today. However, their undeniable qualities are counterbalanced by the fundamental limitations of handmade representations. In this work we show that we can train a single learnable frontend that outperforms mel-filterbanks on a wide range of audio signals, including speech, music, audio events and animal sounds, providing a general-purpose learned frontend for audio classification. To do so, we introduce a new principled, lightweight, fully learnable architecture that can be used as a drop-in replacement of mel-filterbanks. Our system learns all operations of audio features extraction, from filtering to pooling, compression and normalization, and can be integrated into any neural network at a negligible parameter cost. We perform multi-task training on eight diverse audio classification tasks, and show consistent improvements of our model over mel-filterbanks and previous learnable alternatives. Moreover, our system outperforms the current state-of-the-art learnable frontend on Audioset, with orders of magnitude fewer parameters.

## 1 INTRODUCTION

Learning representations by backpropagation in deep neural networks has become the standard in audio understanding, ranging from automatic speech recognition (ASR) (Hinton et al., 2012; Senior et al., 2015) to music information retrieval (Arcas et al., 2017), as well as animal vocalizations (Lostanlen et al., 2018) and audio events (Hershey et al., 2017; Kong et al., 2019). Still, a striking constant along the history of audio classification is the mel-filterbanks, a fixed, hand-engineered representation of sound. Mel-filterbanks first compute a spectrogram, using the squared modulus of the short-term Fourier transform (STFT). Then, the spectrogram is passed through a bank of triangular bandpass filters, spaced on a logarithmic scale (the mel-scale) to replicate the non-linear human perception of pitch (Stevens & Volkmann, 1940). Eventually, the resulting coefficients are passed through a logarithm compression, to replicate our non-linear sensitivity to loudness (Fechner et al., 1966). This approach of drawing inspiration from the human auditory system to design features for machine learning has been historically successful (Davis & Mermelstein, 1980; Mogran et al., 2004). Moreover, decades after the design of mel-filterbanks, Andén & Mallat (2014) showed that they coincidentally exhibit desirable mathematical properties for representation learning, in particular shift-invariance and stability to small deformations. Hence, both from an auditory and a machine learning perspective, mel-filterbanks represent strong audio features.

However, the design of mel-filterbanks is also flawed by biases. First, not only has the mel-scale been revised multiple times (O'Shaughnessy, 1987; Umesh et al., 1999), but also the auditory experiments that led their original design could not be replicated afterwards (Greenwood, 1997). Similarly, better alternatives to log-compression have been proposed, like cubic root for speech enhancement (Lyons & Paliwal, 2008) or 10th root for ASR (Schluter et al., 2007). Moreover, even though matching human perception provides good inductive biases for some application domains, e.g., ASR or music understanding, these biases may also be detrimental, e.g. for tasks that require fine-grained resolution at high frequencies. Finally, the recent history of other fields like computer vision, in which the rise of deep learning methods has allowed learning representations from raw pixels rather than from engineered features (Krizhevsky et al., 2012), inspired us to take the same path.

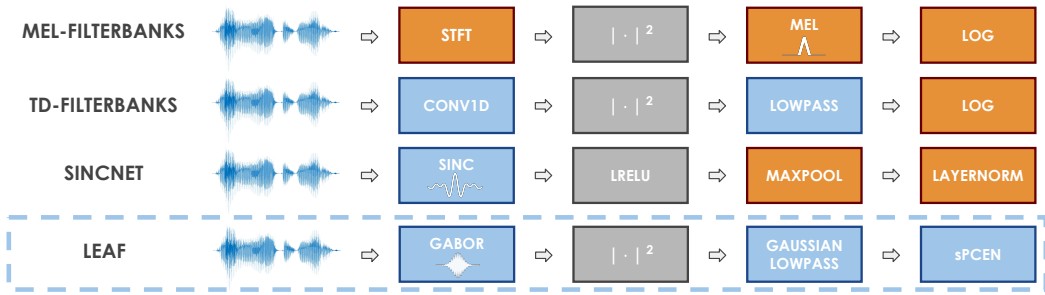

Figure 1: Breakdown of the computation of mel-filterbanks, Time-Domain filterbanks, SincNet, and the proposed LEAF frontend. Orange boxes are fixed, while computations in blue boxes are learnable. Grey boxes represent activation functions.

These observations motivated replacing mel-filterbanks with learnable neural layers, ranging from standard convolutional layers (Palaz et al., 2015) to dilated convolutions (Schneider et al., 2019), as well as structured filters exploiting the characteristics of known filter families, such as Gammatone (Sainath et al., 2015), Gabor (Zeghidour et al., 2018a; Noé et al., 2020), Sinc (Ravanelli & Bengio, 2018; Pariente et al., 2020) or Spline (Balestriero et al., 2018) filters. While tasks such as speech separation have already successfully adopted learnable frontends (Luo & Mesgarani, 2019; Luo et al., 2019), we observe that most state-of-the art approaches for audio classification (Kong et al., 2019), ASR (Synnaeve et al., 2019) and speaker recognition (Villalba et al., 2020) still employ mel-filterbanks as input features, regardless of the backbone architecture.

In this work, we argue that a credible alternative to mel-filterbanks for classification should be evaluated across many tasks, and propose the first extensive study of learnable frontends for audio over a wide and diverse range of audio signals, including speech, music, audio events, and animal sounds. By breaking down mel-filterbanks into three components (filtering, pooling, compression/normalization), we propose LEAF, a novel frontend that is fully learnable in all its operations, while being controlled by just a few hundred parameters. In a multi-task setting over 8 datasets, we show that we can learn a single set of parameters that outperforms mel-filterbanks, as well as previously proposed learnable alternatives. Moreover, these findings are replicated when training a different model for each individual task. We also confirm these results on a challenging, large-scale benchmark: classification on Audioset (Gemmeke et al., 2017). In addition, we show that the general inductive bias of our frontend (i.e., learning bandpass filters, lowpass filtering before downsampling, learning a per-channel compression) is general enough to benefit other systems, and propose a new, improved version of SincNet (Ravanelli & Bengio, 2018). Our code is publicly available[1].

## 2  RELATED WORK

In the last decade, several works addressed the problem of learning the audio frontend, as an alternative to mel-filterbanks. The first notable contributions in this field emerged for ASR, with Jaitly & Hinton (2011) pretraining Restricted Boltzmann Machines from the waveform, and Palaz et al. (2013) training a hybrid DNN-HMM model, replacing mel-filterbanks by several layers of convolution. However, these alternatives, as well as others proposed more recently (Tjandra et al., 2017; Schneider et al., 2019), are composed of many layers, which makes a fair comparison with mel-filterbanks difficult. In the following section, we focus on frontends that provide a lightweight, drop-in replacement to mel-filterbanks, with comparable capacity.

### 2.1  LEARNING FILTERS FROM WAVEFORMS

A first attempt at learning the filters of mel-filterbanks was proposed by Sainath et al. (2013), where a filterbank is initialized using the mel-scale and then learned together with the rest of the network, taking a spectrogram as input. Instead, Sainath et al. (2015) and Hoshen et al. (2015) later

---

[1]https://github.com/google-research/leaf-audio

proposed to learn convolutional filters directly from raw waveforms, initialized with Gammatone filters (Schluter et al., 2007). In the same spirit, Zeghidour et al. (2018a) used the scattering transform approximation of mel-filterbanks (Andén & Mallat, 2014) to propose the time-domain filterbanks, a learnable frontend that approximates mel-filterbanks at initialization and can then be learned without constraints (see Figure 1). More recently, the SincNet (Ravanelli & Bengio, 2018) model was proposed, which computes a convolution with sine cardinal filters, a non-linearity and a max-pooling operator (see Figure 1), as well as a variant using Gabor filters (Noé et al., 2020).

We take inspiration from these works to design a new learnable filtering layer. As detailed in Section 3.1.2, we parametrize a complex-valued filtering layer with Gabor filters. Gabor filters are optimally localized in time and frequency (Gabor, 1946), unlike Sinc filters that require using a window function (Ravanelli & Bengio, 2018). Moreover, unlike Noé et al. (2020), who use complex-valued layers in the rest of the network, we describe in Section 3.1.2 how using a squared modulus not only brings back the signal to the real-valued domain (leading to compatibility with standard architectures), but also performs shift-invariant Hilbert envelope extraction. Zeghidour et al. (2018a) also apply a squared modulus non-linearity, however as described in Section 3.1.1, training unconstrained filters can lead to overfitting and stability issues, which we solve with our approach.

## 2.2 LEARNING THE COMPRESSION AND THE NORMALIZATION

The problem of learning a compression and/or normalization function has received less attention in the past literature. A notable contribution is the Per-Channel Energy Normalization (PCEN) (Wang et al., 2017; Lostanlen et al., 2019), which was originally proposed for keyword spotting, outperforming log-compression. Later, Battenberg et al. (2017) and Lostanlen et al. (2018) confirmed the advantages of PCEN, respectively for large scale ASR and animal bioacoustics. However, these previous works learn a compression on top of fixed mel-filterbanks. Instead, in this work we propose a new version of PCEN and show for the first time that combining learnable filters, learnable pooling, and learnable compression and normalization outperforms all other approaches.

## 3 MODEL

Let $x \in \mathbb{R}^T$ denote a one-dimensional waveform of $T$ samples, available at the sampling frequency $F_s$ [Hz]. We decompose the frontend into a sequence of three components: i) filtering, which passes $x$ through a bank of bandpass filters followed by a non-linearity, operating at the original sampling rate $F_s$; ii) pooling, which decimates the signal to reduce its temporal resolution; iii) compression/normalization, which applies a non-linearity to reduce the dynamic range. Overall, the frontend can be represented as a function $\mathcal{F}_\psi : \mathbb{R}^T \rightarrow \mathbb{R}^{M \times N}$, which maps the input waveform to a 2-dimensional feature space, where $M$ denotes the number of temporal frames (typically $M < T$), $N$ the number of feature channels (which might correspond to frequency bins) and $\psi$ the frontend parameters. The features computed by the frontend are then fed to a model $g_\theta(\cdot)$ parametrized by $\theta$. The frontend and the model parameters are estimated by solving a supervised classification problem:

$$\theta^*, \psi^* = \arg\min_{\theta, \psi} E_{(x,y) \in \mathcal{D}} \, \mathcal{L}(g_\theta(\mathcal{F}_\psi(x)), y), \tag{1}$$

where $(x, y)$ are samples in a labelled dataset $\mathcal{D}$ and $\mathcal{L}$ is a loss function. Our goal is to learn the frontend parameters $\psi$ end-to-end with the model parameters $\theta$. To achieve this, it is necessary to make all the frontend components learnable, so that we can solve the optimization problem in equation 1 with gradient descent. In the following we detail the design choices of each component.

## 3.1 FILTERING

The first block of the learnable frontend takes $x$ as input, and computes a convolution with a bank of complex-valued filters $(\varphi_n)_{n=1..N}$, followed by a squared modulus operator, which brings back its output to the real-valued domain. This convolution step has a stride of 1, therefore keeping the input temporal resolution, and outputs the following time-frequency representation:

$$f_n = |x * \varphi_n|^2 \in \mathbb{R}^T, \quad n = 1, \dots, N, \tag{2}$$

where $\varphi_n \in \mathbb{C}^W$ is a complex-valued 1-D filter of length $W$. It is possible to compute equation 2 without explicitly manipulating complex numbers. As proposed by Zeghidour et al. (2018a), to

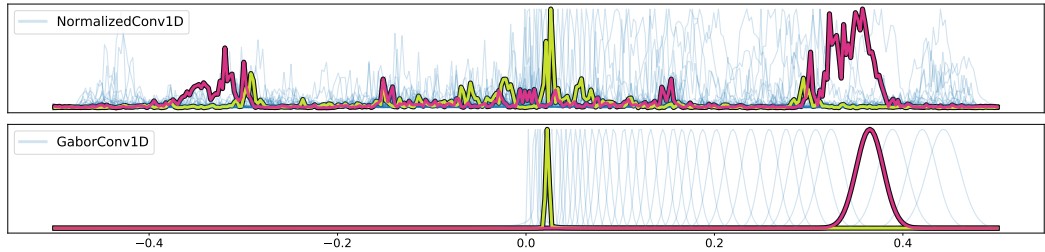

Figure 2: Frequency response of filters at convergence for two parametrizations: normalized 1D filters and Gabor filters, both initialized on a mel scale. We highlight two filters among the 40 of the filterbank: one in yellow in the low frequency range and one in pink at high frequencies.

produce the squared-modulus of a complex-valued convolution with $N$ filters, we compute instead the convolution with $2N$ real-valued filters $\tilde{\varphi}_n, n = 1, \ldots, 2N$, and perform squared $\ell_2$-pooling with size 2 and stride 2 along the channel axis to obtain the squared modulus, using adjacent filters as real and imaginary part of $\varphi_n$. Formally:

$$f_n = |x * \tilde{\varphi}_{2n-1}|^2 + |x * \tilde{\varphi}_{2n}|^2 \in \mathbb{R}^T, \quad n = 1, \ldots, N. \tag{3}$$

We explore two different parametrizations for $\varphi_n$. One relies on standard fully parametrized convolutional filters while the other makes use of learnable Gabor filters.

### 3.1.1 NORMALIZED 1D-CONVOLUTION

Inspired by Zeghidour et al. (2018a), the first version of our filtering component is a standard 1D convolution, initialized with a bank of Gabor filters, which approximates the computation of a mel-filterbank. Thus, at initialization, the output of the frontend is identical to that of a mel-filterbank, but during training the filters can be learned by backpropagation. In our early experiments, we observed several limitations of this approach. First, due to the unconstrained optimization, these filters not only learn to select frequency bands of interest, but also learn a scaling factor. This might lead to instability during training, because the same form of scaling can be also computed by the filter-wise compression component, which is applied at a later stage in the frontend. We address this issue by applying $\ell_2$-normalization to the filter coefficients before computing the convolution. Second, the unconstrained parametrization increases the number of degrees of freedom, making training prone to overfitting. The first panel in Figure 2 shows the frequency response of a bank of $N = 40$ filters, each parametrized with $W = 401$ coefficients. We observe that, at convergence, they are widely spread across the frequency axis, and include negative frequencies. Moreover, the filters are very spiky rather than smooth. To alleviate these issues, in the next section we introduce a parametrization based on a bank of Gabor filters, which reduces the number of parameters to learn, while at the same time enforcing during training a stable and interpretable representation.

### 3.1.2 GABOR 1D-CONVOLUTION

Gabor filters are produced by modulating a Gaussian kernel with a sinusoidal signal. These filters provide several desirable properties. First, they have an optimal trade-off between localization in time and frequency (Gabor, 1946), which makes them a suitable choice for a convolutional network with finite-sized filters. This is in contrast to Sinc filters (Ravanelli & Bengio, 2018), which require using a window function to smooth abrupt variations on each side of the filter. Second, the time and frequency response of Gabor filters have the same functional form, thus leading to interpretable bandpass filters, unlike the unconstrained filters described in the previous section. Finally, Gabor filters are quasi-analytic (i.e., their frequency response is almost zero for negative frequencies) and when combined with the squared modulus the resulting filterbank can be interpreted as a set of subband Hilbert envelopes, which are invariant to small shifts. Due to this desirable property, they have been previously used as (fixed) features for speech and speaker recognition (Falk & Chan, 2009; Thomas et al., 2008). Formally, Gabor filters are parametrized by their center frequencies $(\eta_n)_{n=1..N}$ and inverse bandwidths $(\sigma_n)_{n=1..N}$ as follows:

$$\varphi_n(t) = e^{i2\pi\eta_n t}\frac{1}{\sqrt{2\pi}\sigma_n}e^{-\frac{t^2}{2\sigma_n^2}}, \quad n = 1, \ldots, N, \quad t = -W/2, \ldots, W/2. \tag{4}$$

The frequency response of $\varphi_n$ is a Gaussian centered at frequency $\eta_n$ and of bandwidth $1/\sigma_n$, both expressed in normalized frequency units in $[-1/2, +1/2]$. Therefore, learning these parameters allows learning a bank of smooth, quasi-analytic bandpass filters, with controllable center frequency and bandwidth. In practice, to compute the output of the filtering component, we obtain the impulse response of the Gabor filters $\varphi_n(t)$ over the range $t = -W/2, \ldots, W/2$, and convolve these impulse responses with the input waveform. To ensure stability during training, we clip the center frequencies $(\eta_n)_{n=1..N}$ to be in $[0, 1/2]$, so that they lie in the positive part of the frequency range. We also constrain the bandwidths $(\sigma_n)_{n=1..N}$ in the range $[4\sqrt{2\log 2}, 2W\sqrt{2\log 2}]$, such that the full-width at half-maximum of the frequency response is within $1/W$ and $1/2$.

Gabor filters have significantly fewer parameters than the normalized 1D-convolutions described in Section 3.1.1. $N$ filters of length $W$ are described by $2N$ parameters, $N$ for the center frequencies and $N$ for the bandwidths, against $W \cdot N$ for a standard 1D-convolution. In particular, when using a window length of $25\,\mathrm{ms}$, operating at a sampling rate of $16\,\mathrm{kHz}$, then $W = 401$ samples, and Gabor-based filtering accounts for 200 times fewer parameters than their unconstrained alternatives.

### 3.1.3 TIME-FREQUENCY ANALYSIS AND LEARNABLE FILTERS

A spectrogram, in linear or mel scale, provides an ordered time-frequency representation: adjacent frames represent consecutive time-steps, while frequencies monotonically increase along the feature axis. A learnable frontend that performs filtering by means of convolution with a set of bandpass filters also preserves ordering along the temporal axis. However, the ordering along the feature axis is unconstrained. This can be problematic when applying subsequent operations that rely on frequency ordering. These include, for example: i) operations that leverage local frequency information, e.g., two-dimensional convolutions, which compute a feature representation based on local time-frequency patches; ii) operations that leverage long-range dependencies along the frequency axis, e.g., to capture the harmonic structure of the underlying signal; and iii) augmentation methods that mask adjacent frequency bands, like SpecAugment (Park et al., 2019). To evaluate the impact of enforcing an explicit ordering of the center frequencies of the learned filters, we compared the result of training a frontend using Gabor filters with or without explicitly enforcing ordering of the center frequencies. Interestingly, we observe that even without an explicit constraint, filters that are ordered at the initialization tend to keep the same ordering throughout training, and that enforcing sorted filters has no effect on the performance.

### 3.2 LEARNABLE LOWPASS POOLING

The output of the filtering component has the same temporal resolution as the input waveform. The second step of a learnable frontend is to downsample the output of the filterbank to a lower sampling rate, similarly to what happens in the STFT when computing mel-filterbanks. Previous work relied on max-pooling (Sainath et al., 2015; Ravanelli & Bengio, 2018; Noé et al., 2020), lowpass-filtering (Zeghidour et al., 2018a) or average pooling (Balestriero et al., 2018). Zeghidour et al. (2018b) compare these methods on a speech recognition task and show a systematic improvement when using lowpass filtering instead of max-pooling. More recently, Zhang (2019) showed that in standard 2D convolutional architectures, including ResNet (He et al., 2016) and DenseNet (Huang et al., 2017), a drop-in replacement of max-pooling and average pooling layers with a (fixed) lowpass filter improves the performance for image classification. In the proposed frontend, we extend the pooling layer of Zeghidour et al. (2018a) in two ways. First, while Zeghidour et al. (2018a) adopt a single shared lowpass filter for all input channels, we implement lowpass filtering by means of depthwise convolution, such that each input channel is associated with one lowpass filter. This is useful because each channel in the learnable frontend is characterized by a different bandwidth, and a specific lowpass filter can be learned for each of them. Second, we parametrize these lowpass filters to have a Gaussian impulse response:

$$\phi_n(t) = \frac{1}{\sqrt{2\pi}\sigma_n}e^{-\frac{t^2}{2\sigma_n^2}}, \quad t = -W/2, \ldots, W/2. \tag{5}$$

Note that this is a particular case of Gabor filters with center frequency equal to $0$ and learnable bandwidth. With this choice, we can learn per-channel lowpass pooling functions while adding only $N$ parameters to the frontend model. We initialize all channels with a bandwidth of $0.4$, which gives a frequency response close to the Hann window used by mel-filterbanks.

### 3.3 LEARNING PER-CHANNEL COMPRESSION AND NORMALIZATION

When using a traditional frontend based on a mel-filterbank or STFT, the time-frequency features are typically passed through a logarithmic compression, to replicate the non-linear human perception of loudness (Fechner et al., 1966). The first limitation of this approach is that the same compression function is applied to all frequency bins, regardless of their content. The second limitation is the fixed choice of the non-linearity used by the compression function. While, in most cases, a logarithmic function is used, other compression functions have been proposed and evaluated in the past, including the cubic root (Lyons & Paliwal, 2008) and the 10th root (Schluter et al., 2007). This motivates learning the compression function as part of the model, rather than relying on a handcrafted choice. In particular, Per-Channel Energy Normalization (Wang et al., 2017) was proposed as a learnable alternative to log-compression and mean-variance normalization:

$$\text{PCEN}(\mathcal{F}(t,n)) = \left(\frac{\mathcal{F}(t,n)}{(\varepsilon + \mathcal{M}(t,n))^{\alpha_n}} + \delta_n\right)^{r_n} - \delta_n^{r_n}, \tag{6}$$

where $t = 1, \ldots, M$ denotes the time-step and $n = 1, \ldots, N$ the channel index. In this parametrization, the time-frequency representation $\mathcal{F}$ is first normalized by an exponential moving average of its past values $\mathcal{M}(t,n) = (1-s)\mathcal{M}(t-1,n) + s\mathcal{F}(t,n)$, controlled by a smoothing coefficient $s$ and an exponent $\alpha_n$, with $\varepsilon$ being a small constant used to avoid dividing by zero. An offset $\delta_n$ is then added before applying compression with the exponent $r_n$ (typically in $[0,1]$). Wang et al. (2017) train $\alpha_n$, $\delta_n$, and $r_n$, while treating $s$ as a hyperparameter, or learning a convex combination of exponential moving averages for different, fixed values of $s$. Instead, in this work we learn channel-dependent smoothing coefficients $s_n$, jointly with the rest of the parameters. We call this version sPCEN. This approach was previously used by Schlüter & Lehner (2018) for singing voice detection, except that they did not learn the exponents $\alpha_n$, while we learn all these parameters jointly. Our final frontend cascades a Gabor 1D-convolution, a Gaussian lowpass pooling, and sPCEN. In the rest of the paper, we refer to this model as LEAF, for "LEarnable Audio Frontend".

## 4 EXPERIMENTS

We evaluate our frontend on three supervised learning problems: i) single-task classification; ii) multi-task classification and iii) multi-label classification on Audioset. As baselines, we compare our system to log-compressed mel-filterbanks, learnable Time-Domain filterbanks (Zeghidour et al., 2018a)[2] and SincNet (Ravanelli & Bengio, 2018)[3]. In all our experiments, we keep the same common backbone network architecture, which consists of a frontend, a convolutional encoder, and one (or more) head(s). We adopt the lightweight version of EfficientNet (Tan & Le, 2019) (EfficientNetB0, with $4\,\text{M}$ parameters) as convolutional encoder. On AudioSet, we also experiment with a state-of-the-art CNN14 encoder (Kong et al., 2019), with $81\,\text{M}$ parameters. A *head* is a single linear layer with a number of outputs equal to the number of classes. In the multi-task setting we use a different head for each of the target tasks, all sharing the same encoder.

The input signal sampled at $F_s = 16\,\text{kHz}$ is passed through the frontend which feeds into the convolutional encoder. As baseline, we use a log-compressed mel-filterbank with 40 channels, computed over windows of $25\,\text{ms}$ with a stride of $10\,\text{ms}$. For a fair comparison, both LEAF and the learnable baselines also have $N = 40$ filters, each with $W = 401$ coefficients ($\approx 25\,\text{ms}$ at $16\,\text{kHz}$). The learnable pooling is computed over 401 samples with a stride of 160 samples ($10\,\text{ms}$ at $16\,\text{kHz}$), giving the same output dimension as mel-filterbanks. On AudioSet we use 64 channels instead of 40 as Kong et al. (2019) observed improvements from using 64 mel-filters.

To address the variable length of the input sequences, we train on randomly sampled 1 second windows. We train with ADAM (Kingma & Ba, 2014) and a learning rate of $10^{-4}$ for $1\,\text{M}$ batches, with

---

[2]https://github.com/facebookresearch/tdfbanks
[3]https://github.com/mravanelli/SincNet

Table 1: Test accuracy (%) for single-task classification.

| Task | Mel | TD-fbanks | SincNet | LEAF |
|------|-----|-----------|---------|------|
| Acoustic scenes | $99.2 \pm 0.4$ | $\mathbf{99.5} \pm 0.3$ | $96.7 \pm 0.9$ | $99.1 \pm 0.5$ |
| Birdsong detection | $78.6 \pm 1.0$ | $80.9 \pm 0.9$ | $78.0 \pm 1.0$ | $\mathbf{81.4} \pm 0.9$ |
| Emotion recognition | $49.1 \pm 2.4$ | $57.1 \pm 2.4$ | $44.2 \pm 2.4$ | $\mathbf{57.8} \pm 2.4$ |
| Speaker Id. (VC) | $31.9 \pm 0.7$ | $25.3 \pm 0.7$ | $\mathbf{43.5} \pm 0.8$ | $33.1 \pm 0.7$ |
| Music (instrument) | $\mathbf{72.0} \pm 0.6$ | $70.0 \pm 0.6$ | $70.3 \pm 0.6$ | $\mathbf{72.0} \pm 0.6$ |
| Music (pitch) | $91.5 \pm 0.3$ | $91.3 \pm 0.3$ | $83.8 \pm 0.5$ | $\mathbf{92.0} \pm 0.3$ |
| Speech commands | $92.4 \pm 0.4$ | $87.3 \pm 0.4$ | $89.2 \pm 0.4$ | $\mathbf{93.4} \pm 0.3$ |
| Language Id. | $76.5 \pm 0.4$ | $71.6 \pm 0.5$ | $78.9 \pm 0.4$ | $\mathbf{86.0} \pm 0.4$ |
| Average | $73.9 \pm 0.8$ | $72.9 \pm 0.8$ | $73.1 \pm 0.9$ | $\mathbf{76.9} \pm 0.8$ |

Table 2: Impact of the compression layer on the performance (single task, average accuracy in %).

| | mel-filterbank | | | LEAF- filterbank | | |
|---|---|---|---|---|---|---|
| | log | PCEN | sPCEN | log | PCEN | sPCEN |
| Average | $73.9 \pm 0.8$ | $76.4 \pm 0.8$ | $76.0 \pm 0.7$ | $74.6 \pm 0.7$ | $76.4 \pm 0.8$ | $\mathbf{76.9} \pm 0.8$ |

batch size 256. For Audioset experiments, we train with mixup (Zhang et al., 2017) and SpecAugment (Park et al., 2019). During evaluation, we consider the full-length sequences, splitting them into consecutive non-overlapping 1 second windows and averaging the output logits over windows.

## 4.1 SINGLE-TASK AUDIO CLASSIFICATION

We train independent single-task supervised models on 8 distinct classification problems: acoustic scene classification on TUT (Heittola et al., 2018), birdsong detection (Stowell et al., 2018), emotion recognition on Crema-D (Cao et al., 2014), speaker identification on VoxCeleb (Nagrani et al., 2017), musical instrument and pitch detection on NSynth (Engel et al., 2017), keyword spotting on Speech Commands (Warden, 2018), and language identification on VoxForge (Revay & Teschke, 2019). A summary of the datasets used in our experiments is illustrated in Table A.1.

Table 1 reports the results for each task, with 95% confidence intervals representing the uncertainty due to the limited test sample size. No class rebalancing is applied, neither during training, nor during testing. On average, we observe that LEAF outperforms all alternatives. When considering results for each individual tasks, we observe that LEAF outperforms or matches the accuracy of other frontends, with the notable exception of SincNet on Voxceleb. This is consistent with the fact that SincNet was originally proposed for speaker identification (Ravanelli & Bengio, 2018) and illustrates the importance of evaluating over a wide range of tasks. To evaluate the robustness of our results with respect to the choice of the datasets, we apply a statistical bootstrap (Efron & Tibshirani, 1993) to compute the non-parametric distribution of the difference between the accuracy obtained with LEAF and each of the other frontends, when sampling datasets with replacement among the eight datasets. We test the null hypothesis that the mean of the difference is zero and measure the following one-sided p-values: $p_{\text{Mel}} < 10^{-5}$, $p_{\text{TD-fbanks}} < 10^{-5}$, $p_{\text{SincNet}} = 0.059$. Figure A.1 illustrates the corresponding bootstrap distribution, showing the statistical significance of our results with respect to the choice of the datasets. In Table 2, we study the effect of the compression function. PCEN improves significantly over log-compression on both mel- and learned filterbanks. We also observe that for any choice of the compression function, the learnable frontend matches or outperforms the corresponding mel-filterbank.

## 4.2 MULTI-TASK AUDIO CLASSIFICATION

To learn a general-purpose frontend that generalizes across tasks, we train a multi-task model that uses the same LEAF parameters and encoder for all tasks, with a task-specific linear head. More specifically, a single network with $K$ heads $(h_{\theta_1}, \ldots, h_{\theta_K})$ is trained on mini-batches, uniformly sampled from the $K$ datasets. A mini-batch of size $B$ can be represented as $(x_i^{k_i}, y_i^{k_i})_{i=1..B}$, where $k_i$ is the associated task the example has been sampled from. The multi-task loss function is now computed on a mini-batch as the sum of the individual loss functions:

Table 3: Test accuracy (%) for multi-task classification.

| Task | Mel | TD-fbanks | SincNet | LEAF |
|---|---|---|---|---|
| Acoustic scenes | **99.1** $\pm$ 0.5 | 98.3 $\pm$ 0.6 | 91.0 $\pm$ 1.4 | 98.9 $\pm$ 0.5 |
| Birdsong detection | 81.3 $\pm$ 0.9 | **82.3** $\pm$ 0.9 | 78.8 $\pm$ 0.9 | 81.9 $\pm$ 0.9 |
| Emotion recognition | 24.1 $\pm$ 2.1 | 24.4 $\pm$ 2.1 | 26.2 $\pm$ 2.1 | **31.9** $\pm$ 2.3 |
| Speaker Id. (LBS) | **100.0** $\pm$ 0.0 | **100.0** $\pm$ 0.0 | **100.0** $\pm$ 0.0 | **100.0** $\pm$ 0.0 |
| Music (instrument) | **70.7** $\pm$ 0.6 | 66.3 $\pm$ 0.6 | 67.4 $\pm$ 0.6 | 70.2 $\pm$ 0.6 |
| Music (pitch) | 88.5 $\pm$ 0.4 | 86.4 $\pm$ 0.4 | 81.2 $\pm$ 0.5 | **88.6** $\pm$ 0.4 |
| Speech commands | **93.6** $\pm$ 0.3 | 89.5 $\pm$ 0.4 | 91.4 $\pm$ 0.4 | **93.6** $\pm$ 0.3 |
| Language Id. | 64.9 $\pm$ 0.5 | 58.9 $\pm$ 0.5 | 60.8 $\pm$ 0.5 | **69.6** $\pm$ 0.5 |
| Average | 77.8 $\pm$ 0.7 | 75.8 $\pm$ 0.7 | 74.6 $\pm$ 0.8 | **79.3** $\pm$ 0.7 |

Table 4: Test AUC and d-prime ($\pm$ standard deviation over three runs) on Audioset, with the number of learnable parameters per frontend.

| Frontend | #Params | EfficientNetB0 | | CNN14 (ours) | | CNN14 (Kong et al., 2019) | |
|---|---|---|---|---|---|---|---|
| | | AUC | d-prime | AUC | d-prime | AUC | d-prime |
| Mel | 0 | 0.968 $\pm$ .001 | 2.61 $\pm$ .02 | 0.972 $\pm$ .000 | 2.71 $\pm$ .01 | 0.973 | 2.73 |
| Mel-PCEN | 256 | 0.967 $\pm$ .000 | 2.60 $\pm$ .01 | 0.973 $\pm$ .000 | 2.72 $\pm$ .00 | - | - |
| Wavegram | 300 k | 0.958 $\pm$ .000 | 2.44 $\pm$ .00 | 0.961 $\pm$ .001 | 2.50 $\pm$ .02 | 0.968 | 2.61 |
| TD-fbanks | 51 k | 0.965 $\pm$ .001 | 2.57 $\pm$ .01 | 0.972 $\pm$ .000 | 2.70 $\pm$ .00 | - | - |
| SincNet | 256 | 0.961 $\pm$ .000 | 2.48 $\pm$ .00 | 0.970 $\pm$ .000 | 2.66 $\pm$ .01 | - | - |
| SincNet+ | 448 | 0.966 $\pm$ .002 | 2.58 $\pm$ .04 | 0.973 $\pm$ .001 | 2.71 $\pm$ .01 | - | - |
| LEAF | 448 | 0.968 $\pm$ .001 | 2.63 $\pm$ .01 | **0.974** $\pm$ **.000** | **2.74** $\pm$ **.01** | - | - |

$$\mathcal{L} = \sum_{i=1}^{B} \sum_{k=1}^{K} \mathcal{L}_k \left( h_{\theta_k} \left( g_\theta (\mathcal{F}_\psi(x_i^{k_i})) \right), y_i^{k_i} \right) \delta(k_i, k), \tag{7}$$

where $\theta$ and $\psi$ represent the shared parameters of the encoder and frontend respectively, $\theta_k$, $k = 1, \ldots, K$ the task-specific parameters of the heads $h_{\theta_k}(\cdot)$, and $\delta$ is the Kronecker delta function.

We use the same set of tasks described in Section 4.1 with one exception, replacing VoxCeleb with Librispeech for speaker identification. As illustrated in Table A.1, VoxCeleb has much more classes ($\approx$1200) than any other task. This creates an imbalance in the number of parameters in the heads, making training significantly slower and reducing the accuracy on all other tasks, regardless of the frontend. Table 3 reports the accuracy obtained on each task, for all the frontends. LEAF shows the best overall performance while matching or outperforming all other methods on every tasks. We repeated the bootstrap analysis (Figure A.2) to evaluate the statistical significance of these results with respect to the choice of the datasets and observed the following p-values: $p_{\text{Mel}} = 0.048$, $p_{\text{TD-fbanks}} < 10^{-5}$, $p_{\text{SincNet}} = 10^{-5}$. To the best of our knowledge, it is the first time a learnable frontend is shown to outperform mel-filterbanks over several tasks with a unique parametrization.

## 4.3 MULTI-LABEL CLASSIFICATION ON AUDIOSET

Table 4 shows the performance (averaged over three runs) of the different frontends on Audioset (Gemmeke et al., 2017), a large-scale multi-label dataset of sounds of all categories, described by an ontology of 527 classes. Audioset examples are weakly labelled with one or more positive labels, so we evaluate with the standard metric of this dataset, the balanced d-prime, a non-linear transformation of the AUC averaged uniformly across all classes. EfficientNetB0 trained with LEAF achieves a d-prime of 2.63, outperforming the level achieved when using mel-filterbanks (d-prime: 2.61) or other frontends. We compare this result with the state-of-the-art PANN model (Kong et al., 2019), which reports a d-prime of 2.73 when trained on mel-filterbanks. Kong et al. (2019) also train a learnable "Wavegram" frontend made of 9 layers of 1D- and 2D- convolutions, for a total of 300 k parameters, reporting a d-prime equal to 2.61. EfficientNetB0 on LEAF thus outperforms this system, while having a frontend with 670x fewer parameters, and a considerably smaller encoder. To confirm these results, we also train EfficientNetB0 on the Wavegram and get similar findings.

All these findings are replicated when replacing EfficientNetB0 with our implementation of CNN14: even though the gap between LEAF and baselines reduces as the scores get higher, CNN14 on LEAF matches the current state-of-the-art on AudioSet. To conclude these experiments and show that individual components of LEAF can benefit other frontends, we replace the max-pooling of SincNet with our Gaussian lowpass filter, and LayerNorm (Ba et al., 2016) with the proposed sPCEN. This version, named "SincNet+" in Table 4, significantly outperforms the original version.

### 4.4 ANALYSIS OF LEARNED FILTERS, POOLING AND COMPRESSION

Figure A.3 illustrates the center frequencies learned by the Gabor filtering layer of LEAF on AudioSet, and compares them to those of mel-filterbanks. At a high level, these filters do not deviate much from their mel-scale initialization. On the one hand, this indicates that the mel-scale is a strong initialization, a result consistent with previous work (Sainath et al., 2015; Zeghidour et al., 2018b). On the other hand, there are differences at both ends of the range, with LEAF covering a wider range of frequencies. For example, the lowest frequency filter is centered around 60 Hz, as opposed to 100 Hz for mel-filterbanks. We believe that is one of the reasons that explain the out-performance of LEAF on Audioset, as it focuses on a more appropriate frequency range to represent the underlying audio events. Figure A.4 shows the learned Gaussian lowpass filters at convergence. We see that they all deviate towards a larger frequency bandwidth (characterized by a smaller standard deviation in the time domain), and that each filter has a different bandwidth, which confirms the utility of using depthwise pooling. Figure A.5 shows the learned exponents $r_n$ (initialized at 2.0) and smoothing coefficients $s_n$ (initialized at 0.04) of sPCEN, ordered by filter index. The exponents are spread in $[1.9, 2.6]$, which shows the importance of learning per-channel exponents, as well as the appropriate choice of using a fixed cubic root in previous work (Lyons & Paliwal, 2008), as an alternative to log-compression. Learning per-channel smoothing coefficients also allows deviating from the initialization. Interestingly, most filters keep a low coefficient (slowly moving average), except for the filter with the highest frequency, which has a significantly faster moving average ($s \approx 0.15$).

### 4.5 ROBUSTNESS TO NOISE

We compare the robustness of LEAF and mel-filterbanks to synthetic noise, on the Speech Commands dataset. To do so, we artificially add Gaussian noise to the waveforms both during training and evaluation, with different gains to obtain a Signal-to-Noise Ratio (SNR) from $+\inf$ (no noise) to $-5\,\mathrm{dB}$. Figure A.6 shows that while performance (averaged over three runs) degrades for all models as SNR decreases, LEAF is more resilient than mel-filterbanks. In particular, when using a logarithmic compression, the Gabor 1-D convolution and Gaussian pooling of LEAF maintain a significantly higher accuracy than mel-filterbanks. Using PCEN has an even higher impact on performance when using mel-filterbanks, the best results being obtained with LEAF + PCEN.

## 5 CONCLUSION

In this work we introduce LEAF, a fully learnable frontend for audio classification as an alternative to using handcrafted mel-filterbanks. We demonstrate over a large range of tasks that our model is a good drop-in replacement to these features with no adjustment to the task at hand, and can even learn a single set of parameters for general-purpose audio classification while outperforming previously proposed learnable frontends. In future work, we will move yet a step forward in removing handcrafted biases from the model. In particular, our model still relies on an underlying convolutional architecture, with fixed filter length and stride. Learning these important parameters directly from data would allow for an easier generalization across tasks with various sampling rates and frequency content. Moreover, we believe that the general principle of learning to filter, pool and compress can benefit the analysis of non-audio signals, such as seismic data or physiological recordings.

## 6 ACKNOWLEDGMENTS

Authors thank Dick Lyon, Vincent Lostanlen, Matt Harvey, and Alex Park for helpful discussions. Authors are also thankful to Julie Thomas for helping with the design of Figure 1. Finally, authors thank the reviewers of ICLR 2021 for their feedback that helped improving the quality of this work.

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

## A APPENDIX

Table A.1: Datasets used in the experiments. Default train/test splits are always adopted.

| Task | Name | Classes | Train examples | Test examples |
|------|------|---------|----------------|---------------|
| Audio events | Audioset | 527 | 1,832,720 | 17,695 |
| Acoustic scenes | TUT Urban 2018 | 10 | 7,829 | 810 |
| Birdsong detection | DCASE2018 | 2 | 32,129 | 3,561 |
| Emotion recognition | Crema-D | 6 | 5,146 | 820 |
| Speaker Id. | Voxceleb | 1,251 | 138,361 | 8,249 |
| Speaker Id. | Librispeech | 251 | 25,740 | 2,799 |
| Music (instrument) | Nsynth | 11 | 289,205 | 12,678 |
| Music (pitch) | Nsynth | 128 | 289,205 | 12,678 |
| Speech commands | Speech commands | 35 | 84,771 | 10,700 |
| Language Id. | Voxforge | 6 | 126,610 | 18,378 |

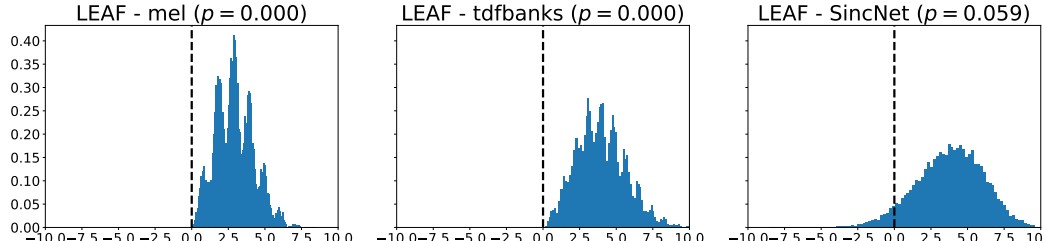

Figure A.1: Distribution of the difference between the accuracy (%) obtained with LEAF and each of the other frontends when sampling 8 datasets with replacement, in the single task setting.

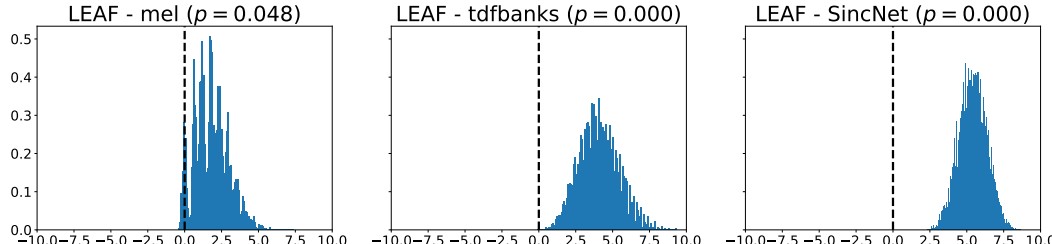

Figure A.2: Distribution of the difference between the accuracy obtained with LEAF and each of the other frontends when sampling 8 datasets with replacement, in the multi-task setting.

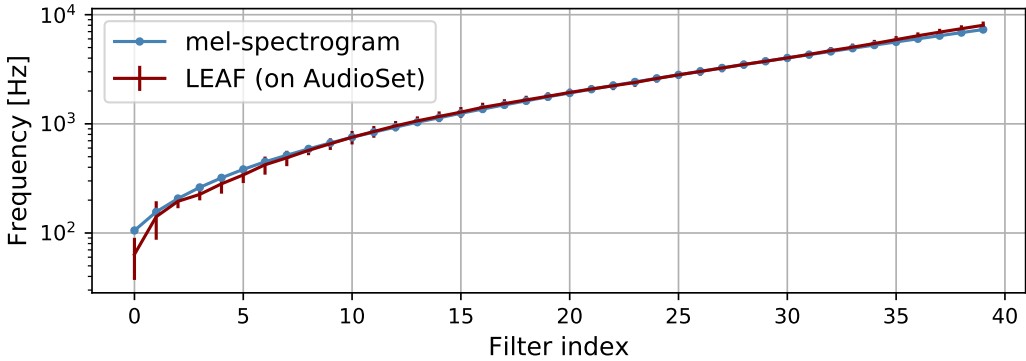

Figure A.3: Comparison between the filters learned by LEAF and the mel scale.

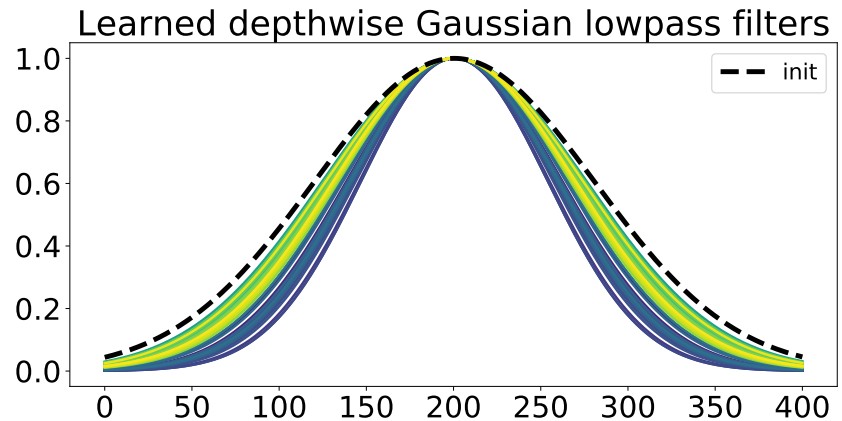

Figure A.4: Learned Gaussian lowpass filters of LEAF on AudioSet. The dotted line represents the initialization, identical for all filters.

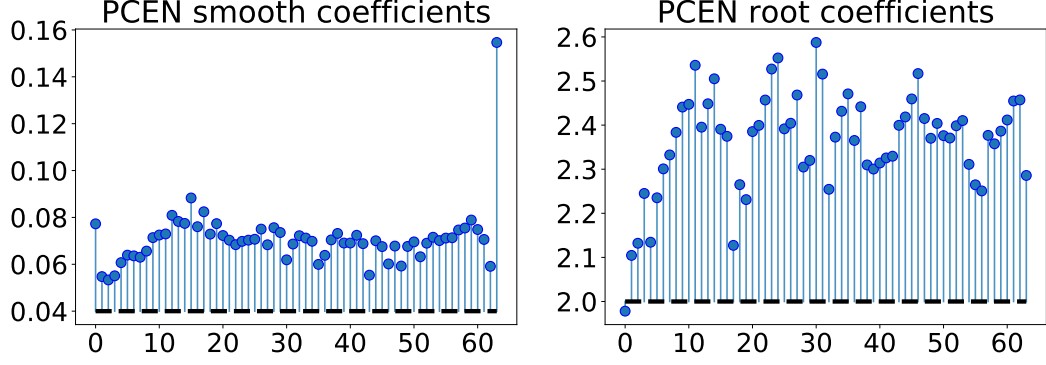

Figure A.5: Values of learned sPCEN parameters of LEAF trained on AudioSet. The dotted lines represent the initialization, identical for all filters.

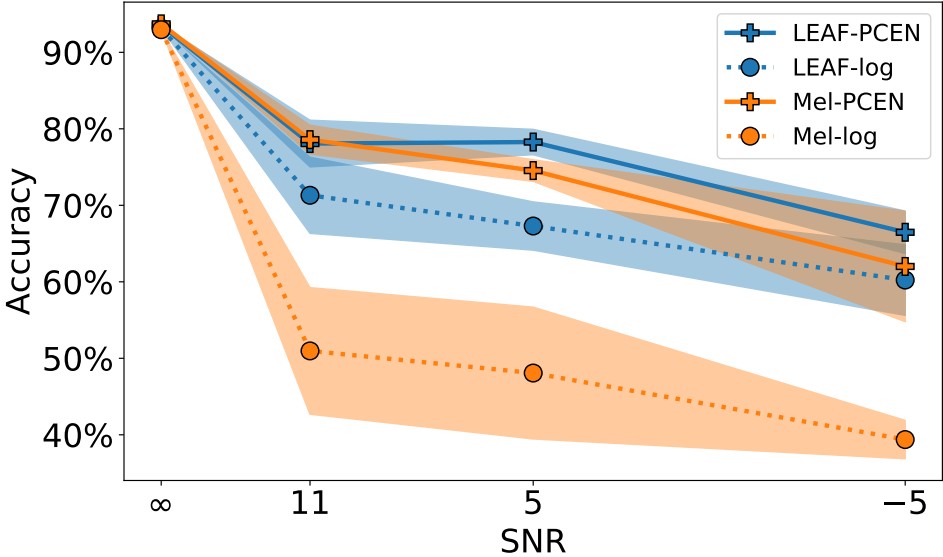

Figure A.6: Test accuracy (%) on the test set of the Speech Commands dataset with varying SNR.

