# OpenReview forum: "LEAF: A Learnable Frontend for Audio Classification"
_ICLR.cc/2021/Conference — ICLR 2021 Poster_

### Official Review · AnonReviewer2 · 2020-10-28
**Review for A Universal Learnable Audio Frontend**

**Rating:** 4
**Confidence:** 4

**Review:**

In this work, the authors introduce a learnable front-end (LEAF) for audio. The paper evaluates it on several tasks in the audio domain such as acoustic event classification, speaker identification, keyword spotting, language identification, music classification etc. The study reports that proposed features outperform or perform similar to Mel filterbanks for many tasks. This paper is well written and easy to follow. The references are excellent as well. I have summarized my comments below which will help in improving the quality of this manuscript:

1. While the authors evaluate the proposed front-end on several tasks, the evaluation protocols are missing. Especially, the training partition and test partition and number of segments in each class. For example, did they calculate accuracy using balanced classes? Was classification done at the frame level or the utterance level? I would highly recommend authors to add the experiment protocol to the manuscript as Table 1A does not provide enough information.

2. The authors evaluate LEAF on several tasks however the model architecture for classification models remains constant. Typically Mel FBs are utilized across architectures and one of the major shortcomings of the proposed work is the integration of LEAF with state-of-the-art models (for example x-vectors for SID). This would give more understanding about the generalization capability of the LEAF.

3. As pointed out by authors in the introduction, Mel FBs are invariant to deformation and noise-robust. All the audio problems addressed in this work encounter varying levels of noise in the real-world. It is extremely important to assess the noise robustness of the proposed front-end. I would recommend the authors test LEAF with different levels of noise.

4. For results in Table 1 the statistical significance should be computed. For the acoustic scene task, the difference in performance is very less and this dataset only has 810 samples overall for testing. Therefore, without confidence intervals, it is very hard to conclude the performance of LEAF.

Minor comments:
1. Please add this missing reference:
Davis, S. and Mermelstein, P., 1980. Comparison of parametric representations for monosyllabic word recognition in continuously spoken sentences. IEEE transactions on acoustics, speech, and signal processing, 28(4), pp.357-366.

2. In Table 1, Mel and LEAF has the same accuracy for Music instrument and needs to be bolded.

3. mel ---> Mel

4. In Fig 2, you may improve the font size of x-axis labels.

---

> ### Author Response · Authors · 2020-11-17
> **Response to AnonReviewer2**
>
> We thank the reviewer for their feedback.
>
> ====”While the authors evaluate the proposed front-end on several tasks, the evaluation protocols are missing. Especially, the training partition and test partition and number of segments in each class.”
>
> We included additional details about the evaluation protocol in the revised manuscript. Table A.1 now reports the number of train/test examples and the number of distinct classes for each dataset. Note that we used the default train/test partitions that are specified by each dataset.
>
> ====”For example, did they calculate accuracy using balanced classes?”
>
> No class rebalancing was performed, neither during training, nor during evaluation. A note has been inserted in the text.
>
> ====”Was classification done at the frame level or the utterance level?”
>
> This is already specified in the last paragraph of Section 4:
> “To address the variable length of the input sequences, we train on randomly sampled 1 second windows….During evaluation, we consider the full-length sequences, splitting them into consecutive non-overlapping 1 second windows and averaging the output logits over windows”
>
> ====”The authors evaluate LEAF on several tasks however the model architecture for classification models remains constant. Typically Mel FBs are utilized across architectures and one of the major shortcomings of the proposed work is the integration of LEAF with state-of-the-art models (for example x-vectors for SID). This would give more understanding about the generalization capability of the LEAF.”
>
> We repeated our experiments with different model architectures (EfficientNetB0 and CNN14) to demonstrate that the properties of the learnable frontend do not depend on the specific architecture used to process the computed audio features. Table 4 reports these new results.
> In this paper we focused on audio classification tasks and we revised the title, introduction  and conclusion accordingly. At the same time, we acknowledge that LEAF can be potentially useful in other contexts (e.g., x-vector for SID) and we will definitely explore this interesting direction in our future work. Moreover, we will release an open-source implementation of LEAF, along with our implementation of CNN14, TD-fbanks and SincNet, to foster application of our model to other tasks.
>
> ====”As pointed out by authors in the introduction, Mel FBs are invariant to deformation and noise-robust. All the audio problems addressed in this work encounter varying levels of noise in the real-world. It is extremely important to assess the noise robustness of the proposed front-end. I would recommend the authors test LEAF with different levels of noise.”
>
> We definitely acknowledge that the robustness to noise is an important aspect when deploying models in the real world. When evaluating LEAF, we were faced with two options: a) artificially add noise to clean input samples; b) use a wide variety of datasets, some of which also include different forms of noise. A shortcoming of option a) is that it requires to arbitrarily define the noise distribution along multiple axes (additive/convolutional, stationary/non-stationary, etc.). In our original submission we opted for b), including in our evaluation set datasets like Acoustic scenes, Speech commands, Birdsong detection and especially Audioset, which do contain noise sampled from a real distribution.
> In addition, in the revised paper, we have now added a further experiment on the SpeechCommands dataset, with additive Gaussian noise. These experiments show that LEAF is at least as robust as mel-filterbanks when using PCEN, and significantly more robust when using logarithmic compression (see Figure A.1).
>
> ====”For results in Table 1 the statistical significance should be computed. For the acoustic scene task, the difference in performance is very less and this dataset only has 810 samples overall for testing. Therefore, without confidence intervals, it is very hard to conclude the performance of LEAF.”
>
> We amended Table 1 and Table 3, reporting next to each accuracy value the confidence interval capturing the limited sample size.
> While for some of the datasets other frontends might outperform LEAF (e.g., TD-Fbanks for Acoustic Scenes or SincNet for SpeakerID), we are interested in the average performance across a diverse set of datasets. To evaluate the robustness of our results to the specific choice of the datasets, we applied the statistical bootstrap, resampling with replacement a set of K = 8 datasets, and computing the non-parametric distribution of DeltaAccuracy = Accuracy_LEAF - Accuracy_X (where X in {Mel, TD-FBanks, SincNet}). We tested the null hypothesis that the mean of DeltaAccuracy is zero, and measured a p-value equal to, respectively <1e-5, <1e-5 and 0.059, thus demonstrating the statistical significance of the LEAF outperformance. The corresponding bootstrap distribution is illustrated in Appendix.

---

### Official Review · AnonReviewer1 · 2020-10-28
**Worthwhile investigation but lack of humility and truthfulness**

**Rating:** 8
**Confidence:** 5

**Review:**

This paper presents a new learnable representation fo audio signal classification and compares it to the classical mel-filterbanks representation and two other learnable representations on a broad range of audio classification tasks, from birdsongs to pitch, instrument, language or emotion recognition. The proposed representation combines several parameterized representation techniques from the recent litterature. It is reported to yield on par or better classification results than the other methods on several of these tasks using single- or multi-task learning.

Pros:
- Learning an ultimate, universal, generic representation for all audio signals that renders the 80 years old mel-frequency scale obsolete is certainly an attractive goal
- The proposed representation carefully and elegantly combines the best parts of several recently proposed parameterized representations and enjoys a nice interpretability while requiring few parameters to learn.
- Comparing different audio representations on such a broad range of audio classification task is a welcome and unmatched effort, to the best of my knowledge.

Cons:
- The paper lacks humility in its story-telling and its style. It employs formulations such as "lived through the history of audio", or "challenging the historical statu quo" when refereing to mel-frequency representations, although by the authors' own admition in the paper, a large amount of research effort has already been given in recent years towards learnable audio representations (the authors cite a dozen papers but there are more). Hence, this paper is not a first attempt. And despite the pompous use of "universal" in the title, I believe it is not a last attempt either. The authors claim that the proposed representation "outperform mel-filterbanks over several tasks with a unique parametrization" but this is far from clear when looking at the results carefully. In the majority of the tasks, the representation performs either slightly worse, equal, or about 0.5% better than Mel-filterbanks. It is not clear whether such improvement is significant, since no error bar or standard deviation is provided in the results (a sadly common habit in the audio litterature). The only tasks where a truly significant improvement is reported are language identification and emotion recognition, which are also the tasks where all the methods perform the poorest. It looks like any significant difference between the 4 compared approaches would vanish if these two tasks were omitted. The reason why the proposed representation performs well on these two very specific tasks is not clear and not discussed.
- More generally, the paper would be much more valuable if it gave a sense of WHAT is actually learned by the proposed method. Is the final representation significantly different from a mel-filterbank? Given how close to mel most reported results are, this is doubtful. In fact, Fig. A.1. strongly suggests that LEAF just re-learned mel, but strangely this figure is never commented. Some comments on the learned compression-parameters would also be appreciated.
- At least one important comparison point is clearly missing in the reported results: STFT + PCEN or mel-filterbanks + PCEN, e.g., Wang et al. (2017) or Schlüter & Lehner (2018) [note that the latter already uses sPCEN rather than PCEN, contrary to the authors' claim] . Omitting this from the comparisons prevents one from knowing whether the proposed parameterized Gabor filterbank brings any advantage over another time-frequency representation like STFT or mel-filterbanks. Less critically, another missing comparison point is LEAF + CNN14, in Table 4.
- What the authors refer to as "audio" in the title and throughout the paper is in fact much more narrow, namely "audio classification". Learnable audio representations have been studied in a broader context in recent years, e.g., speech enhancement, source separation, dereverberation, sound localization or audio (re-)synthesis. In fact, one of the important breakthroughs recently brought by learnable audio frontends was in source separation with the paper TasNet (Luo et al. 2018) which is not cited by the authors. In the same context, (Ditter and Gerkmann 2020) presented a learnable gammatone-like filterbank and showed that fully-parameterized learned filterbanks tended to have logarithmic spread in frequencies. Moreover, the use of learnable analytical filterbanks/Hilbert pairs due to their envelop extraction/shift invariant properties was already discussed in depth in (Pariente et al. 2019) [cited in the paper].

Overall, while comparing different learnable audio representations on a broad range of audio classification tasks is a timely and worthwhile topic, and while the proposed representation elegantly combines several recent ideas in this area, the general presentation and angle of the paper strongly lacks humility. Instead of the proposed title, something like "Benchmark of learnable audio representations on a broad range of classification tasks" would be more truthful to the work. To make the investigation more worthwhile and insightful, additional comparison points (STFT + PCEN, mel-frequency + PCEN, Gabor + log, etc.) as well as an analysis of what the model has actually learned would be needed.

======= Review edit after authors' revisions ======
The changes made by the authors in the title, abstract, introduction and conclusion to narrow the scope of the paper, better contextualize it, and make it more humble and truthful are very welcome. The extra experiments, figures, addition of error bars and new statistical tests are also a real plus. In doing so, the authors addressed all of my major concerns.

For these reasons, changed my evaluation score from 5 to 8.

---

> ### Author Response · Authors · 2020-11-17
> **Response to AnonReviewer1 (Part 1)**
>
> We thank the reviewer for their feedback. We split our response in two comments due to the 5000 characters limitation.
>
> ====”The paper embarassingly lacks humility in its story-telling and its style. It employs formulations such as "lived through the history of audio", or "challenging the historical statu quo" when refereing to mel-frequency representations, although by the authors' own admition in the paper, a large amount of research effort has already been given in recent years towards learnable audio representations (the authors cite a dozen papers but there are more). Hence, this paper is not a first attempt. And despite the pompous use of "universal" in the title, I believe it is not a last attempt either.”
>
> Our work is motivated by the observation that despite the large corpus of work on learnable frontends, of which we cite a significant part, state-of-the-art classification models still rely on mel-filterbanks. We do acknowledge this vast amount of prior work in the introduction and related work, and we build on top of these previous approaches to propose an alternative. This won’t indeed be a last attempt at replacing mel-filterbanks, however we believe that the broad range of tasks we consider, our focus on combining and improving the best components of previous work, as well as open-sourcing all our models makes this contribution valuable to the audio classification community. We revised the title, abstract, introduction and conclusion of the paper to more precisely frame the scope of the paper, which focuses on audio classification as correctly pointed out by the reviewer, as well as reformulated our claims.
>
> ====”The authors claim that the proposed representation "outperform mel-filterbanks over several tasks with a unique parametrization" but this is far from clear when looking at the results carefully. In the majority of the tasks, the representation performs either slightly worse, equal, or about 0.5% better than Mel-filterbanks. It is not clear whether such improvement is significant, since no error bar or standard deviation is provided in the results (a sadly common habit in the audio litterature). The only tasks where a truly significant improvement is reported are language identification and emotion recognition, which are also the tasks where all the methods perform the poorest. It looks like any significant difference between the 4 compared approaches would vanish if these two tasks were omitted. The reason why the proposed representation performs well on these two very specific tasks is not clear and not discussed.”
>
> We revised Table 1 and Table 3, reporting the values of the confidence intervals to reflect the uncertainty due to the limited sample size in each evaluation dataset.
> While for some of the datasets other frontends might outperform LEAF (e.g., TD-Fbanks for Acoustic Scenes or SincNet for SpeakerID), we are interested in the average performance across a diverse set of datasets. To evaluate the robustness of our results to the specific choice of the datasets, we applied the statistical bootstrap, resampling with replacement a set of K = 8 datasets, and computing the non-parametric distribution of DeltaAccuracy = Accuracy_LEAF - Accuracy_X (where X in {Mel, TD-FBanks, SincNet}). We tested the null hypothesis that the mean of DeltaAccuracy is zero, and measured a p-value equal to, respectively <1e-5, <1e-5 and 0.056 for single-task classification and 0.048, <1e-5 and <1e-5 for multi-task classification, thus demonstrating the statistical significance of the LEAF outperformance.
>
> ===”More generally, the paper would be much more valuable if it gave a sense of WHAT is actually learned by the proposed method. Is the final representation significantly different from a mel-filterbank? Given how close to mel most reported results are, this is doubtful. In fact, Fig. A.1. strongly suggests that LEAF just re-learned mel, but strangely this figure is never commented. Some comments on the learned compression-parameters would also be appreciated.”
>
> We added a Section 4.4 of analysis which comments this figure as well as adds visualizations and analysis of learned Gaussian lowpass filters and PCEN parameters, thanks to the additional space allowed at this stage. As correctly observed by the reviewer, at a high level, these filters do not deviate much from their mel-scale initialization. On the one hand, this indicates that the mel-scale is a strong initialization, a result consistent with previous work (references added in the paper). On the other hand, there are differences at both ends of the range, with LEAF covering a wider range of frequencies. For example, the lowest frequency filter is centered around 60Hz, as opposed to 100Hz for mel-filterbanks. We believe that is one of the reasons that explain the out-performance of LEAF on Audioset, as it focuses on a more appropriate frequency range to represent the underlying audio events.

---

> > ### Author Response · Authors · 2020-11-17
> > **Response to AnonReviewer1 (Part 2)**
> >
> > ====”At least one important comparison point is clearly missing in the reported results: STFT + PCEN or mel-filterbanks + PCEN, e.g., Wang et al. (2017) or Schlüter & Lehner (2018) [note that the latter already uses sPCEN rather than PCEN, contrary to the authors' claim] . Omitting this from the comparisons prevents one from knowing whether the proposed parameterized Gabor filterbank brings any advantage over another time-frequency representation like STFT or mel-filterbanks. Less critically, another missing comparison point is LEAF + CNN14, in Table 4.”
> >
> > Table 2 already shows an ablation study of the compression function in the single task setting, including mel-filterbanks + PCEN. We furthermore added mel-filterbanks + PCEN in our AudioSet experiments (Table 4). Moreover, we reimplemented CNN14 and added the results to Table 4, showing that LEAF still maintains an advantage over learnable frontends and mel-filterbanks.
> >
> > ====”[note that the latter already uses sPCEN rather than PCEN, contrary to the authors' claim]”
> >
> > Thanks for pointing out this mistake, we corrected our reference to this work.
> >
> > ====”What the authors refer to as "audio" in the title and throughout the paper is in fact much more narrow, namely "audio classification". Learnable audio representations have been studied in a broader context in recent years, e.g., speech enhancement, source separation, dereverberation, sound localization or audio (re-)synthesis. In fact, one of the important breakthroughs recently brought by learnable audio frontends was in source separation with the paper TasNet (Luo et al. 2018) which is not cited by the authors. In the same context, (Ditter and Gerkmann 2020) presented a learnable gammatone-like filterbank and showed that fully-parameterized learned filterbanks tended to have logarithmic spread in frequencies. Moreover, the use of learnable analytical filterbanks/Hilbert pairs due to their envelop extraction/shift invariant properties was already discussed in depth in (Pariente et al. 2019) [cited in the paper].”
> >
> > We agree that our contribution is narrower than what the title suggests.  We acknowledge that the state-of-the-art in source separation has used neural networks on the raw waveform rather than spectrogram masking for several years (e.g. TasNet, DPRNN). We added these references in the introduction. In this work we are interested in alternatives to mel-filterbanks in tasks where they are still used in state-of-the-art systems, namely audio classification. We modified the title and the description of our contribution accordingly.

---

### Official Review · AnonReviewer4 · 2020-10-29
**The paper well connects the relationship between hand-crafted audio frontends (mel-spectrogram) with learnable frontends.**

**Rating:** 7
**Confidence:** 4

**Review:**

The paper shows a detailed interpretation on the relationship between each component of hand-crafted audio front-ends (such as mel-spectrograms) and learnable counterparts. To do that, they followed the narratives presented from the previous works such as SincNet and improved the model by changing the several components of it. The authors grouped the audio front-ends into mainly three parts which are filtering, pooling, and compression. And, the contributions were made at each stage. For filtering stage, instead of learning all the parameters of the convolution layer, they let the model to learn only center frequency and bandwidth of the filterbanks that are initially assigned with Gabor filters. For pooling stage, instead of using simple average or max poolings, they let the model to learn low pass filtering with small parameters. For compression, instead of using log based dynamic compression, they extended Per-Channel Energy Normalization by replacing a fixed smoothing factor to learnable parameters and named it to sPCEN.

To evaluate the proposed approach, they evaluated the models on 8 audio classification tasks which might have diverse audio and label characteristics (such as acoustic scene sound, animal sound, music, speech). The compared models are mainly mel-spectrogram and SincNet. The results shows that the proposed model outperforms the comparisons for most tasks. Then, they further ran a multi-task classification experiment to obtain universal audio front-ends. And, the results show the proposed learnable front-ends is showing some generalization ability on most tasks. Finally, they evaluated the proposed model on large-scale audio classification dataset (AudioSet) and verified that the proposed front-ends is also showing the good performance on it.

In page 4, the authors mentioned that the l2 normalization helps distinguishing the role of filtering and compression. I think this contribution is not trivial, so if the authors can add more experiments (or plots) to show the difference between models with and without l2 normalization, then it would be helpful.

The backbone model used in the paper is fixed, and showing that the proposed audio front-ends shows similar trends with multiple backend models can verify better generalization ability of the proposed approach. So, if the authors can add additional experiments with multiple backends, then it would be helpful.

---

> ### Author Response · Authors · 2020-11-17
> **Response to AnonReviewer4**
>
> We thank the reviewer for their feedback.
>
> ====”In page 4, the authors mentioned that the l2 normalization helps distinguishing the role of filtering and compression. I think this contribution is not trivial, so if the authors can add more experiments (or plots) to show the difference between models with and without l2 normalization, then it would be helpful.”
>
> When the l2 normalization of the filters is disabled, we observed that the filters tend to have heterogeneous gains. This is problematic, especially when a non-linear compression operation is applied to the output of filtering (e.g., via PCEN), since the local slope and curvature of the nonlinearity depend on the scale of its input. Hence, we argue that normalization should always be applied. Also, note that all our results use Gabor 1D convolution kernels (as it worked consistently better in our early experiments), which are normalized by construction. The impact of the compression layer is illustrated in Table 2.
>
> ====”The backbone model used in the paper is fixed, and showing that the proposed audio front-ends shows similar trends with multiple backend models can verify better generalization ability of the proposed approach. So, if the authors can add additional experiments with multiple backends, then it would be helpful.”
>
> In the revised version of our paper we experiment with different architectures to verify the generalization properties of LEAF. Namely, we use the CNN14 backbone network recently proposed in [Kong et al., 2019] as it achieves state-of-the-art results on Audioset. Our additional results reported in an updated Table 4 show that the trend is similar regardless of the backend encoder.

---

### Official Review · AnonReviewer3 · 2020-10-31
**A simple but interesting frontend for audio classification tasks**

**Rating:** 7
**Confidence:** 4

**Review:**

The paper proposes a learnable frontend for classification tasks on audio signals. The proposed learnable audio frontend (LEAF) is a generalization of a mel filterbank, used commonly in machine audition.
LEAF consists of a Gabor filterbank, magnitude-squared nonlinearity, Gaussian lowpass filter and previously-proposed per-channel energy normalization. Learnable parameters of LEAF are the center frequencies and bandwidths of Gabor filters, bandwith of lowpass Gaussian filters and smoothing coefficients of sPCEN. The proposed LEAF matches the performance of competing frontends in most of the cases and leads to some improvements in some cases.
In general, I like the paper and the proposed frontend is very sensible from perspective of audio-related applications.
However, I believe there’s some exaggeration in terms of impact of the proposed frontend based on the results in this paper. The proposed system is not actually challenging status quo, as many learnable frontends have been proposed in the literature in the past (as also listed in the references).
The paper is easy to read and authors communicate their contribution clearly.
However, I believe that the title may be somewhat too general: LEAF is evaluated only on classification tasks, and IMHO that should be indicated in the title as well. There are other tasks, such as speech enhancement, where LEAF-like frontend may work well, but that is out of scope of this paper.
Experimental results show that LEAF is performing well in the considered tasks. However, it would be interesting to understand the differences in performance when the encoder & head are changed and/or increased.
More specifically, the current results are obtained using a lightweight EfficientNet and linear heads. Is there any particular reason for this setup? Would the conclusions change with a different encoder/head?
Also, understanding of the influence of the filterbank setup (number of channels, window length, stride) would be beneficial.

Details:
(1) Title should reflect the fact that LEAF has been evaluated only on classification tasks
(2) Abstract: “over a wide range of audio domains” —> It would be more appropriate to talk about a range of applications in audio domain.
(3) Abstract: “unprecedented” -> I believe this is a bit exaggerated
(4) Introduction: “this might not be the optimal approach for non-human sounds” —> This is a strange argument, and I believe the authors are confusing sound perception and sound production. Human sound perception works quite well for recognition on non-human sounds. The authors imply that a system which replicates a system mimicking human perception is suboptimal for non-human sounds. However, human auditory system is not designed for processing of only human-made sounds. Furthermore, optimality depends on the application, so stating that something is not optimal for a class of sounds makes no sense without the defined application, which in this case could be recognition of “acoustic events or animal vocalizations”
(5) Introduction, last paragraph: “wide and diverse range of tasks, including speech, music, audio events, and animal vocalizations” —> Signals, such as speech or music are not tasks. A task can be, e.g., speech recognition.
(6) Conclusion: Stating there’s a “historical statu quo of using hand-crafted mel-filterbanks” with so many end-to-end systems giving the best performance in different applications is a bit too much.


======= Review edit after authors' revisions ======

Most of my concerns have been resolved in the significantly-improved revised version of the paper.

---

> ### Author Response · Authors · 2020-11-17
> **Response to AnonReviewer3**
>
> We thank the reviewer for their feedback.
>
> ====”The proposed system is not actually challenging status quo, as many learnable frontends have been proposed in the literature in the past (as also listed in the references).”
>
> We acknowledge in the introduction that this problem has been addressed many times in the recent years. However, despite this huge corpus of work on replacing mel-filterbanks, we observe that state-of-the-art systems for audio classification, ASR or speaker recognition (reference added in the introduction) still rely on mel-filterbanks. This shows that learnable frontends have not been widely adopted for discriminative tasks, while they have become standard for tasks such as speech separation (references added in the introduction as well). The motivation for our work is both to benchmark previously proposed methods across many audio classification tasks, as well as proposing a convincing alternative, that we hope will contribute to the adoption of learnable frontends as an alternative to mel-filterbanks.
>
> ====”However, I believe that the title may be somewhat too general: LEAF is evaluated only on classification tasks, and IMHO that should be indicated in the title as well.”
>
> We revised the title, abstract and introduction of the paper to more precisely frame the scope of the paper, which focuses on audio classification as correctly pointed out by the reviewer.
>
> ====”However, it would be interesting to understand the differences in performance when the encoder & head are changed and/or increased.”
>
> In our terminology, a “head” is solely the last classification layer (which is different per-task in the multi-task setting). Any additional layer is part of the “encoder”. We acknowledge that in the multi-task setting one could increase the depth of the “per-task” network section from one layer to several, but to reduce the hyperparameter search to a reasonable space we relied on one head. However, we agree that showing results with different encoders is important as mel-filterbanks work well across a broad range of architectures. As a consequence, we added results with the state-of-the-art CNN14 on AudioSet in Table 4. Despite the significant change in architecture scale, we still observe improvements from using LEAF over mel-filterbanks and other learnable frontends.
>
> ====”Also, understanding of the influence of the filterbank setup (number of channels, window length, stride) would be beneficial.”
>
> In our single and multi-task experiments we use the standard number of 40 filters, however when training on AudioSet we use 64 channels instead of 40 as Kong et al. found improvements from using 64 filters (added in the paper). To reduce the space of models to explore we stick to the standard 25ms window with 10ms hop size, the most commonly used parameters in the literature. However, as described in the conclusion, in future work we want to address the limitation of choosing a fixed window size and stride, shared among all filters regardless of their frequency bandwidth, which comes from casting the Gabor filtering as a 1-D convolution layer.
>
> ===”Human sound perception works quite well for recognition on non-human sounds. The authors imply that a system which replicates a system mimicking human perception is suboptimal for non-human sounds. However, human auditory system is not designed for processing of only human-made sounds.“
>
> Thanks for this remark, we agree that our statement was imprecise. Indeed, the human auditory system is not designed for processing human-made sounds only, and the interaction between perception and production are out of the scope of this paper. We also agree that “optimality” depends on the task, and that for a single class of signal (e.g. speech) the best bank of filters will likely vary with the task (e.g. ASR, speaker identification or paralinguistic classification). We edited this part.
>
> ===="(2) Abstract: “over a wide range of audio domains” [...] (3) Abstract: “unprecedented” -> I believe this is a bit exaggerated"
>
> We reformulated the abstract accordingly.
>
> ====”(5) [...] Signals, such as speech or music are not tasks. A task can be, e.g., speech recognition.”
>
> This terminology was indeed imprecise, we corrected it by mentioning “diverse range of audio signals”.
>
> ====”Conclusion: Stating there’s a “historical statu quo of using hand-crafted mel-filterbanks” with so many end-to-end systems giving the best performance in different applications is a bit too much”
>
> As described in the introduction, most “end-to-end” systems for audio classification or ASR still use handcrafted features, and learnable frontends are yet to be integrated into state-of-the-art systems for these tasks. This is unlike tasks such as speech separation for which training end-to-end systems from the waveform has become the standard. We revised the title, scope and claims of the paper, as well as added references to show the successful application of learnable frontends to speech separation.

---

### Author Response · Authors · 2020-11-17
**Response to all reviewers**

We thank the reviewers for their feedback and suggestions. We found them very useful to improve the paper and support our claims more convincingly. We summarize here the main changes applied to the paper and replied to all individual comments below.

* We revised the title, abstract, introduction and conclusion of the paper to more precisely frame its scope, which focuses on audio classification as correctly pointed out by AnonReviewer1 and AnonReviewer3.
* We revised Table 1 and Table 3, reporting the values of the confidence intervals to reflect the uncertainty due to the limited sample size in each evaluation dataset, as pointed by AnonReviewer1 and AnonReviewer2. We also performed a statistical bootstrap analysis to measure uncertainty with respect to the choice of datasets. These analyses confirm the advantage of LEAF w.r.t. Mel-filterbanks, Time-Domain filterbanks and SincNet.
* We added experiments on AudioSet with a state-of-the-art “CNN14” encoder, to address the concern of all reviewers regarding the use of a single encoder. This architecture allows us to match the current state-of-the-art on this dataset when using the LEAF frontend. We will include our reimplementation of CNN14 in the open-source release of LEAF, along with our implementations of Time-Domain Filterbanks and SincNet.
* While the Mel+PCEN baseline was already included in our single task experiments over several datasets, we also included it in our experiments on AudioSet, for both encoders, thanks to the suggestion of AnonReviewer1. On this dataset, the performance is identical to Mel+Log.
* To address the concern of AnonReviewer2 we added experiments on speech commands with additive Gaussian noise at varying Signal-to-Noise ratio. These experiments, reported in Section 4.5 and Figure A.6 show the robustness of LEAF w.r.t. mel-filterbanks, both when using a log-compression and PCEN.
* We added an analysis of the learned filters, Gaussian lowpass and PCEN parameters in Section 4.4 and Figures A3-5, as suggested by AnonReviewer1.

Overall, we did our best to address thoroughly all main concerns raised by the reviewers and hope these changes will be taken in consideration.

---

### Comment · ~Sanghyuk_Chun1 · 2021-03-13
**Thanks for your work**

Hi, thanks for the wonderful work!

I really like your paper's motivation: using learnable filterbanks initialized by the mel filterbanks.
We published a paper at ICASSP'20, **"Data-driven Harmonic Filters for Audio Representation Learning"** from the same motivation.

We did propose learnable filterbanks motivated by harmonicity of the audio domain and demonstrated the effectiveness on music-tagging, keyword spotting, and sound event detection benchmarks.
Could you consider our work as one of the prior works of yours?

Anyway, thanks for your nice paper!


Won, Minz, et al. "Data-driven harmonic filters for audio representation learning." ICASSP 2020-2020 IEEE International Conference on Acoustics, Speech and Signal Processing (ICASSP). IEEE, 2020.

---

### Decision · Program_Chairs · 2021-01-07
**Final Decision**

**Decision:**

Accept (Poster)

**Comment:**

All Reviewers agree that the paper has a clear and solid contribution. Furthermore, all of them highlight that the paper has improved significantly after revision. Hence, my recommendation is to ACCEPT the paper. As a brief summary, I highlight below some pros and cons that arose during the review and meta-review processes.

Pros:
- Comparison across network architectures.
- Comparison across a broad range of different data sets.
- Compactness of the representation (few parameters to learn).
- Authors will share code.

Cons:
- Role of L2 normalization could be further discussed.